# Complex Networks Analyses of Antibiofilm Peptides: An Emerging Tool for Next-Generation Antimicrobials’ Discovery

**DOI:** 10.3390/antibiotics12040747

**Published:** 2023-04-13

**Authors:** Guillermin Agüero-Chapin, Agostinho Antunes, José R. Mora, Noel Pérez, Ernesto Contreras-Torres, José R. Valdes-Martini, Felix Martinez-Rios, Cesar H. Zambrano, Yovani Marrero-Ponce

**Affiliations:** 1CIIMAR/CIMAR, Interdisciplinary Centre of Marine and Environmental Research, University of Porto, 4450-208 Porto, Portugal; aantunes@ciimar.up.pt; 2Department of Biology, Faculty of Sciences, University of Porto, 4169-007 Porto, Portugal; 3Universidad San Francisco de Quito (USFQ), Colegio de Ciencias e Ingenierías “El Politécnico”, Instituto de Simulación Computacional (ISC-USFQ), Diego de Robles y vía Interoceánica, Quito 170157, Pichincha, Ecuador; jrmora@usfq.edu.ec (J.R.M.);; 4Universidad San Francisco de Quito (USFQ), Grupo de Medicina Molecular y Traslacional (MeM&T), Colegio de Ciencias de la Salud (COCSA), Escuela de Medicina, Edificio de Especialidades Médicas and Instituto de Simulación Computacional (ISC-USFQ), Diego de Robles y vía Interoceánica, Quito 170157, Pichincha, Ecuador; econtrerastorres88@gmail.com; 5Undoso Consulting, Miami, FL 33185, USA; 6Facultad de Ingeniería, Universidad Panamericana, Augusto Rodin No. 498, Insurgentes Mixcoac, Benito Juárez, Ciudad de México 03920, Mexico; felix.martinez@up.edu.mx; 7Departamento de Ciencias de la Computación, Centro de Investigación Científica y de Educación Superior de Ensenada (CICESE), Ensenada 22860, Baja California, Mexico

**Keywords:** antibiofilm peptide, chemical space, StarPep toolbox, complex network, centrality measure, motif discovery

## Abstract

Microbial biofilms cause several environmental and industrial issues, even affecting human health. Although they have long represented a threat due to their resistance to antibiotics, there are currently no approved antibiofilm agents for clinical treatments. The multi-functionality of antimicrobial peptides (AMPs), including their antibiofilm activity and their potential to target multiple microbes, has motivated the synthesis of AMPs and their relatives for developing antibiofilm agents for clinical purposes. Antibiofilm peptides (ABFPs) have been organized in databases that have allowed the building of prediction tools which have assisted in the discovery/design of new antibiofilm agents. However, the complex network approach has not yet been explored as an assistant tool for this aim. Herein, a kind of similarity network called the half-space proximal network (HSPN) is applied to represent/analyze the chemical space of ABFPs, aiming to identify privileged scaffolds for the development of next-generation antimicrobials that are able to target both planktonic and biofilm microbial forms. Such analyses also considered the metadata associated with the ABFPs, such as origin, other activities, targets, etc., in which the relationships were projected by multilayer networks called metadata networks (METNs). From the complex networks’ mining, a reduced but informative set of 66 ABFPs was extracted, representing the original antibiofilm space. This subset contained the most central to atypical ABFPs, some of them having the desired properties for developing next-generation antimicrobials. Therefore, this subset is advisable for assisting the search for/design of both new antibiofilms and antimicrobial agents. The provided ABFP motifs list, discovered within the HSPN communities, is also useful for the same purpose.

## 1. Introduction

Microorganism biofilms have aroused increasing interest within the scientific community [1,2]. Biofilms are microbial communities composed of either single or multiple species that can be adhered to different types of surface, thereby allowing their survival in changing environmental conditions caused by the use of biocides and antibiotics in industrial and clinical settings, as well as by other circumstances such as the presence of UV light, heavy metals, anaerobic conditions, salinity, pH gradients, etc. [3] However, they frequently cause environmental and industrial issues due to their adherence to metals, river rocks, deep-sea vents, and plant tissues; they have had a negative impact on human health by settling in body tissues and medical implant materials [3,4].

According to the National Institute of Health (NIH), biofilms are responsible for more than the 65% of microbial infections in humans, and for 80–90% of chronic conditions, representing a serious healthcare issue [5,6]. Thus, chronic inflammation, pain and damage to certain tissues or organ systems caused by biofilm-associated infections may be manifested in endocarditis, cystic fibrosis, urinary tract infections, and periodontitis [1]. Biofilms are especially hard to remove from implanted medical devices, e.g., catheters, stents, prosthetic heart valves, pacemakers, and artificial joints or limbs [7,8]. Furthermore, when planktonic forms (free-living microbes) are detached from biofilms, they can trigger other complications for patients, e.g., bacteremia, thromboembolism and septic episodes [9,10]. Globally, the prevalence of biofilm-associated multi-drug resistant (MDR) infections among hospitalized patients ranges from 17.9% to 100.0%, and these are mostly caused by bacteria such as *S. aureus*, *A. baumannii*, *K. pneumoniae*, *P. aeruginosa* and *E. coli*, which can also be found as MDR free-living strains. As a consequence, only 98,000 deaths attributed to them annually in the United States, according to the US Centre for Disease Control and Prevention [11].

Although microbial biofilms have long represented a threat to human health due to their resistance to antibiotics [12,13], there are currently no approved antibiofilm agents for clinical treatments, despite years of research [5]. There are only two antibiofilm candidates in the pipeline of clinical trials: (*i*) nitric oxide, a known regulator of biofilms [14], has been proposed for treating chronic rhinosinusitis (Phase 2, NCT04163978), and (*ii*) the human monoclonal antibody (TRL1068) against the bacterial protein DNABII, which stabilizes DNA in the extracellular matrix of biofilms [15], is being evaluated for treating prosthetic joint infections (Phase 1, NCT04763759) [5]. Currently, exploration of biofilms’ formation, mainly through OMICS approaches, has uncovered new genetic and protein entities as targets for biofilm modulation. Knowledge on molecular signaling in biofilms’ formation has paved the way for exploration of natural and synthetic peptides to target such new entities [5]. For example, the nucleotide second-messenger guanosine tetraphosphate and pentaphosphate ((p)ppGpp) is found in all bacteria as part of the stringent stress response, also playing an important role in biofilm formation in many species [16,17]. Therefore, (p)ppGpp is an excellent target for developing antibiofilm agents, and surprisingly, cationic amphipathic peptides related to the antimicrobial and host defense function can bind directly to (p)ppGpp for its degradation [5,16]. Other promising second-messenger-like target in bacteria is the cyclic diguanylate (c-di-GMP), which regulates a wide range of cellular functions from biofilm formation to growth and survival. In this sense, a c-di-GMP-sequestering peptide (CSP) was designed considering the high binding ability of the c-di-GMP with one of its protein effectors, a CheY-like (Cle) protein found in *Caulobacter crescentus.* The CSP was developed from a short arginine-rich region located at the C termini of Cle [5,18]. Thus, synthetic peptides derived from reported antimicrobial peptides (AMPs) have been also explored to find more promising antibiofilm agents/targets for clinics [5,19]. As a consequence, several derivatives from registered AMPs such as LL-37 and Indolicidin, from human and bovine origin, respectively, have been synthetized based on the potent antibiofilm/antibacterial actions of their natural templates [20].

This background points out AMPs and their synthetic relatives as promising alternatives for developing antibiofilm agents, also leveraging the multi-functionality of the AMPs class through their immunomodulatory, anti-inflammatory, wound-healing, antifungal and antibacterial activities, especially their potentialities to target MDR planktonic strains [21,22]. It is also important to highlight the relative low cost of peptides’ production in comparison with other emerging therapeutic alternatives [23]. Thus, their antibiofilm activity has gained attention in the AMPs world, being evaluated in vitro and in vivo models and registered either in general AMP databases such as UniProtKB [24], the Antimicrobial Peptide Database (APD) [25], the Data Repository of Antimicrobial Peptides (DRAMP) [26], or in others exclusively dedicated to antibiofilm peptides, such as the Biofilm-active AMPs (BaAMPS) database [27]. This last specific database has been useful for researchers to train machine learning (ML) models to predict antibiofilm activity [28,29,30,31].

Despite efforts to populate AMPs databases and build ML models to assist the discovery/design of antibiofilm peptide drugs, the success rate has been extremely low, considering there is no antibiofilm agent in clinical use [5]. Therefore, the development of new in silico approaches, providing new insights into the classical drug discovery pipeline, is always welcome. Thus, emerging tools and databases based on graph and networks science could be integrated in the discovery/design pipeline of peptide drugs through analyzing the chemical space of bioactive peptides [32,33,34]. Consequently, the relevant features of antibiofilm peptides (ABFPs) can be approached by applying complex networks to identify privileged scaffolds and motifs, aiming to develop next-generation antimicrobials able to combat MDR microbes by either targeting their biofilm or planktonic forms. Namely, complex networks can be useful for (i) analyzing the structural diversity of ABFPs according to the network topology/modularity, (ii) selecting the most representative and atypical ABFPs, determined mainly by centrality measures, (iii) identifying relevant functionalities (other than the antibiofilm contributing to efficient antimicrobial activity) by weighting their associated metadata with centrality measures, (iv) identifying ABFP motifs within networks communities, and (v) determining of a reduced set of peptides that actually represents the original antibiofilm chemical space.

Here, the half-space proximal network (HSPN) was applied to represent the structural (sequence-based) space of anti-biofilm peptides registered in the graph-based database (StarPepDB) [35], which probably has the most comprehensive subset of antibiofilm peptides collected from both generic AMPs databases and specific databases for antibiofilms (Available online: http://mobiosd-hub.com/starpep/ (accessed on 5 January 2023)). The HSPN is a local proximity graph that extracts a low-degree spanner of the complete graph. Each node is associated with its nearest neighbor, clearing from the complete graph all redundant nodes in the nearest neighbor’s direction by using a half-space hyperplane [36,37]. HSPNs do not consider all the possible pairwise relationships between nodes; instead, these networks apply the half-space proximal test over the set of nodes, obtaining a connected network with a smaller fraction of the maximum number of edges. As they show a lower density, a lesser RAM memory is needed for their construction; therefore, they were recently applied by our group to study the chemical space of antiparasitic peptides [34].

The estimation of several HSPN parameters, e.g., number of communities, centrality measures, metadata associated with central and atypical nodes, and neighborhood analysis, along with network visual mining, has allowed effective exploration of the chemical space of antibiofilm peptides, aiming to assist in the discovery/design of next-generation antimicrobials targeting both planktonic and biofilm microbial forms. Such a new approach was made possible by mainly using network science tools such as the StarPep toolbox [35] and Gephi [38], complemented with classical bioinformatic analyses.

## 2. Materials and Methods

### 2.1. Half-Space Proximal Network Building 

The building of half-space proximal networks (HSPNs) was described in [34,36,37]. Here, they were applied to represent the chemical/structural space of 174 antibiofilm peptides (ABFPs) which were registered in StarPepDB after removing redundant peptides at 98% similarity from the original set of 221 ABFPs; this was achieved by applying the Smith–Waterman algorithm [39]. Thus, the resulting 174 ABFPs were used to generate HSPNs. In doing so, an optimized set of alignment-free (AF) sequence descriptors found by feature selection methods were used to represent ABFPs as network nodes [36]. The Euclidean distance metrics with min-max normalization were applied to determine the pairwise similarity relationships among them. ABFPs within the HSPN are clustered by using the modularity optimization algorithm based on the Louvain method [40]; thus, peptide communities sharing similar features can be displayed in the network. All these steps were performed using the StarPep toolbox (zip distribution) associated with StarPepDB version 1.0 (CAMD-BIR International network, Quito, Ecuador, available online: http://mobiosd-hub.com/starpep/ (accessed on 5 January 2023)) [35].

### 2.2. Metadata Networks

Metadata networks (METNs) were also explained in [36]. They are another type of network that do not display similarity relationship among the nodes/peptides, similar to the HSPNs. METNs are bi-layer networks represented by the ABFPs and their associated metadata (origin, database, function, target), respectively [35]. Particularly, one layer is composed of the 174 ABFPs, and the other by their sourcing database and origin. Therefore, ABFPs belonging from the same database are connected to the same “database node”, but additionally can be connected to nodes representing other databases. METNs can include other layers (metadata) and multi-type links so that hierarchical connections may be set. METNs are also constructed using the StarPep toolbox v1.0. They aim to gain a better insight into the associated data (antimicrobial activity on MDR strains, immunomodulatory, anti-inflammatory, etc.) from the ABFPs in order to select privileged scaffolds for the development of next-generation antimicrobial agents.

### 2.3. Networks Similarity Cutoff Analysis

The optimal similarity threshold for the HSPN was set up by analyzing the network parameters/topology behavior at varying the pairwise similarity cutoffs between nodes (peptides). HSPNs representing the ABFPs were built with the Euclidean distance metric, but changing similarity threshold from 0 (no cutoff) to 0.95. The exploration was performed from the minimum cutoff 0, where all possible similarity relationships (edges) are displayed, increasing to 0.3, 0.4, etc. until 0.95; only edges with similarity values higher than the applied threshold are shown in the network.

The resulting HSPNs at each similarity cutoff are exported as GraphML files from StarPep toolbox (v1.0-zip distribution) to be imported in Gephi v0.10 [38]. Gephi allows the calculation of a comprehensive set of network parameters not implemented in StarPep toolbox v1.0, such as average degree, density, modularity, average clustering coefficient (ACC), number of communities and singletons, network diameter, and average path length at each similarity threshold. The joint analysis of all these network properties together with the visual mining of the resulting network topologies was considered to determine the optimal value (Appendix A). The optimal cutoff should draw the most informative network topology that best represents/models the ABF chemical space. These exploratory analyses will be addressed in detail in the Results and Discussion sections.

### 2.4. Network Visualization 

Network visualization is conducted in Gephi v0.10 by applying layout algorithms to the original HSPNs, allowing a readable network representation wherein nodes are intended to not overlap [38]. The Fruchterman–Reingold algorithm was applied to layout HSPNs as a circle wherein communities are represented by different colors, while the nodes size was scaled according to centralities measures [41]. Several types of centrality measures were calculated for each node: (*i*) node degree, (*ii*) harmonic, (*iii*) betweenness and (*iv*) hub-bridge. For HSPNs, all these centrality measures were calculated, while for the networks with metadata associated with peptides (e.g., database, origin, target and function), betweenness is the centrality of choice. 

These networks can also be directly shown by using the principal components as the cartesian coordinates of nodes (peptides) represented by a set of AF molecular descriptors. In this case, no layout algorithm is applied.

### 2.5. Scaffold Extraction by Centrality Measures

The HSPN model selected from the similarity threshold analysis was used to retrieve the most central and atypical ABFPs from its communities and singletons. The ABFPs were ranked down separately according to the harmonic and hub-bridge centrality measures, then a process of redundancy reduction was performed using the Scaffold extraction plugin from the StarPep toolbox v1.0 [36]. This procedure aims to avoid an overrepresentation of the ABFPs according to their centrality values by applying different sequence identity cutoffs from 0.90 to 0.30. In this way, in a peptide pair bearing similar centrality measures, only the one with the highest centrality is retained when sharing with its competing peptide’s identity above a predefined cutoff, e.g., 0.90. Thus, several non-redundant and representative sets of ABFPs corresponding to the cutoffs of 0.90, 0.80, 0.70, 0.50, 0.45, 0.40 and 0.30 were extracted. The identity measure was calculated by the Smith–Waterman local alignment algorithm using default settings [39]. As this is a pairwise iterative comparison, the resulting extracted sets contain peptides bearing unique values of centrality measures and sharing sequence identities under the predefined cutoffs.

### 2.6. Selection of the Most Representative Extracted Subset

As several subsets were extracted by applying both the harmonic and hub-bridge centralities at different identity cutoffs (0.9–0.3), the selection of the subset optimally representing the original HSPN space with a minimum number of ABFPs per metric is needed. This selection was performed by visual mining from the HSPN overlapping of the extracted subsets on the HSPN model. The evaluated subsets represented between 30–45% of the original HSPN space (174 ABFPs). We considered for the selection the spatial distribution (spatial coverage) of the reduced subset on the HSPN model, represented by the cartesian coordinates nodes estimated from the two most relevant principal components. Once the subsets corresponding to the best overlap with the minimum identity cutoff and the smallest number of peptides were found for each metric, the union of the centralities (HC ∪ HB) was also explored at 0.45, 040 and 0.35 identity thresholds. The subsets from the union of two centralities should have more coverage than when a single one is applied.

### 2.7. Motif Discovery

#### 2.7.1. Multiple Sequence Alignments

The motif discovery process was conducted on the ABFPs communities from the HSPN model. A total of 11 communities were analysed; 8 of them showed more than 2 members (clusters 4, 7, 9, 11, 14, 15, 17 and 22), while clusters 6 and 28 only contained 2 peptides. The 20 singletons were gathered in one cluster. The detection of anti-biofilm motifs using alignments was approached as follows:-Communities with more than 2 ABFPs, including the one containing 20 singletons, were aligned independently using multiple sequence alignment (MSA) algorithms. The algorithms of choice were MAFFT (Multiple Alignment using Fast Fourier Transform) v7.487 with the iterative refinement FFT-NS-i option [42] and MUSCLE (Multiple Sequence Comparison by Log- Expectation) v3.8 [43], publicly available at https://www.ebi.ac.uk/Tools/msa/, EMBL-EBI, Cambridgeshire, UK, accessed on 22 December 2022.-The conserved motifs were detected by jointly analyzing the consensus sequences and Seq2Logo, implemented in the Jalview v2.11.2.5 program [44] and EMBOSS Cons v6.6.0, available at https://www.ebi.ac.uk/Tools/msa/emboss_cons, EMBL-EBI, Cambridgeshire, UK, accessed on 22 December 2022).-Communities with only 2 ABFPs were pairwise aligned using local and global alignment algorithms [39,45]. The resulting alignments were also imported to the Jalview v2.11.2.5 program [44] to estimate the consensus for the detection of the motifs.

#### 2.7.2. Alignment-Free (AF) Detection

The AF detection of motifs was carried out similarly by communities. They were analyzed in STREME v5.5.1 (Sensitive, Thorough, Rapid, Enriched Motif Elicitation) to discover ungapped motifs that are enriched with respect to a control set generated by shuffling input peptides (https://meme-suite.org/meme/tools/streme, MEME Suite, Washington, DC, USA, accessed on 3 January 2023) [46]. The analyses were performed via MEME suite v5.5.1 [47]. The motif width was set between 3–5 amino acids length. STREME applies a statistical test at *p*-value threshold = 0.05 to determine the enrichment of motifs in the input peptides compared to the control set as a stopping criterion.

#### 2.7.3. Motif Enrichment Analysis

Simple Enrichment Analysis (SEA) v5.5.1 (https://meme-suite.org/meme/tools/sea, MEME Suite, Washington, DC, USA, accessed on 16 January 2023) [48] from the MEME suite v5.5.1 was used to validate the motif discovery process by evaluating the enrichment of each listed motif in external benchmark datasets of ABFPs. The relative enrichment ratio of each motif in the query vs. control sequences is defined as
(1)$$Ratio=(\left(TP+1\right)/\left(NPOS+1\right))∕\left(\left(FP+1\right)/\left(NNEG+1\right)\right),$$
where *NPOS* is the number of positive sequences (query peptides) in the input, and *NNEG* is the number of control sequences in the input.

### 2.8. Overall Workflow Integrating Complex Networks to Next-Generation Antimicrobials Development

The overall workflow integrating complex network analyses as an emerging tool in the next-generation antimicrobials’ development pipeline from ABFPs is displayed in Figure 1. The figure summarizes the methodological steps previously described to achieve the proposed goal as follows: HSPN building, HSPN visualization by delineating its communities and scaling the node size according to centrality values, the selection of an HSPN model (optimal cutoff) using a similarity cutoff analysis, and the selection of a representative subset of ABFPs using the scaffold extraction algorithm from the HSPN model.

The metadata associated with the representative ABFP subset were analyzed by METNs visual mining to identify privileged scaffolds which have shown promising antimicrobial activities on MDR strains, together with antibiofilm action but non-toxicity toward mammalian cells. In turn, the HSPN model (optimal similarity cutoff) was used to discover ABFP motifs within the network communities using alignment-based (AB) and AF tools (MSA, STREME, SEA). The ABFP motifs enriched in two characterized datasets will also be useful for designing next-generation antimicrobials (Figure 1).

## 3. Results and Discussion

### 3.1. Half-Space Proximal Network Model

HSPNs of ABFPs were built with the Euclidean distance metric, but changing similarity threshold from 0.0 to 0.95. As long as the cutoff increases, the network density decreases due to the loss of edges satisfying the similarity cutoff. Therefore, several nodes become disconnected from the giant component of the network, and appear as isolated communities or singletons representing atypical sequences. The original HSPNs (with no similarity cutoff) have the particularity that all nodes are fully connected, as in the giant components of half-space graphs; however, upon applying increased similarity cutoffs, increasingly sparser graphs with unconnected nodes (singletons) are displayed.

Then, some network parameters were retrieved at different similarity cutoffs in order to determine the optimal parameter for determining the most informative network topology. In relevance order, network parameters such as density, modularity, average clustering coefficient and number of communities were analyzed (Figure 2). 

The network density is the actual number of edges over the maximum number of possible edges in a network. If the density is too high, the understanding of network topological features becomes complicated, and if it is too low, then get loses useful information; therefore, a compromise between both extremes is needed. Generally, a network density of around 0.1 is acceptable; however, HSPNs are sparser networks showing much lower density values (<0.01). HSPNs are non-classical similarity networks because their node connections are determined by a predefined half-space proximal (HSP) test that generates a connected network but that only uses a small fraction of the maximum number of possible edges. More detailed information about the HSPNs construction can be found at [36]. Considering they are sparser networks by definition, upon increasing the similarity cutoff, HSPNs’ density tends to drop even more, only retaining a reduced number of edges weighted with high similarity.

The modularity of the networks was also analyzed at each similarity cutoff. This is a network parameter that compares the density within a community with the expected density for the same group of nodes in a random network. We calculated modularity and the number of communities using the modularity optimization clustering algorithm (based on the Louvain method [40]). Unlike the network density, both the modularity and the number of communities/singletons are significantly increased, especially upon applying similarity cutoffs from 0.5 (Figure 2).

On the other hand, the average clustering coefficient (ACC) is a global measure of nodes’ neighborhood connectivity and can also be used for evaluating network topology changes against similarity thresholds. Although at a similarity cutoff of 0.80 all network parameters displayed a dramatical change (Figure 2), the optimal value was selected by jointly analyzing all HSPN parameters and topologies (Appendix A). The point at which the network density dramatically drops but at the same time the modularity increases (while considering the trade-off between the number of communities/singletons generated) is a good starting criterion for selection. In this sense, the cutoff value should be between 0.6–0.7, where the density and modularity have an inverse behavior, while the ACC does not suffer any dramatic change and the number of communities and singletons are reasonable in order to display an informative network topology (Figure 2). Thus, the optimal cutoff of 0.65 was selected by analyzing all the HSPN parameters displayed in Figure 1 and Appendix A, and the networks’ topologies as well. The selected HSPN (cutoff = 0.65) was the best network, projecting the antibiofilm chemical space and grouping the ABFPs in a reasonable number of both interconnected and isolated communities, in which their peptides probably share common or singular physicochemical and biological properties. The relevance of this HSPN will be addressed from now on.

In addition, the degree distributions of the HSPN with no similarity cutoff and at 0.65 were plotted to explore the behavior of these networks as generic models [49]. The node degree distribution of HSPN with no cutoff shows a bell-shaped distribution around a maximum value of 8 (Figure 3 and Appendix A). This bell-shaped distribution is inherent from the random models. However, when the optimal similarity cutoff is applied, the maximum value is reached at node degree 3, and the classical bell-shaped distribution is lost. Instead, several small bell-shaped patterns with peaks appear within the node degree intervals. Therefore, the HSPN model evidently does not display random behavior, which indicates that it could be used as a topological network model (Figure 3 and Appendix A).

### 3.2. Network Visual Mining

#### 3.2.1. Visual Mining of HSPNs; The Most Central and Atypical ABFPs

In addition to the numerical characterization of HSPNs by the calculation of network parameters, their visualization also provides new and simple insights to unravel the complex relationships of the objects they represent. In our case, the HSPNs were used to represent and analyze the chemical space of 174 non-redundant ABFPs by applying AF similarity networks with the application of the half-proximal space graphs [34,36,37]. Network visualization can mirror several network parameters such as density and communities; node size can be ranked according to different centrality measures, e.g., node degree, harmonic, hub-bridge, betweenness, etc. Thus, the most important or central peptides can be highlighted, and the edges weighted with high or low similarities. This work aimed at the outset to exploit the visual representation of complex networks representing ABFPs in order to analyze their structural space and associated metadata, both of which are relevant for the discovery and design of antibiofilm agents [31]. 

Both the original HSPN (no cutoff) and the HSPN model (cutoff at 0.65) representing the structural space occupied by the 174 ABFPs are depicted in Figure 4, and the Appendix A complements the network projection through numerical characterization. Networks communities are highlighted with different colors, and nodes’ importance is represented by the node degree centrality. The original HSPN shows 5 communities clearly identified by different colors, while in the HSPN model bearing 30 communities (20 singletons included), their delineation by color is more difficult (see details in Appendix A). However, since the HSPN model is a low-density network with a smaller number of edges (325) than the original HSPN (689), it allows for a better depiction of the nodes’ relationships (Appendix A).

The most relevant nodes according to the node degree were clearly identified in the HSPN with no cutoff (e.g., starPep_07526, starPep_02281, starPep_00048, starPep_03668, starPep_08958), whereas upon applying the similarity cutoff, a significant fraction of graph edges are lost and the nodes’ degree decreases. This is why only two of the most connected nodes were highlighted (the starPep_00048 and starPep_03668). As these two peptides were also brought up in the original HSPN in similar locations and clusters, it can be deduced that changes in the HSPN topology upon applying similarity cutoffs do not alter the most popular peptides in both networks. In order to address this question, the top ten most relevant peptides according to each four centrality measures (node degree, harmonic, betweenness and hub-bridge) were extracted from the HSPNs with and without similarity cutoff (Appendix A). Then, the intersection of the resulting four peptide sets was analyzed for each HSPN. Table 1 displays the common peptides identified from the ten top-ranked ABFPs by four, three and two centrality measures, from HSPNs with and without similarity cutoff. The last four rows of each HSPN represent singular peptides identified for each of the four centralities. The cluster/community containing the ten top-ranked ABFPs is also displayed.

Table 1 shows a different composition of common and singular peptides from the ten top-ranked ABFPs by centrality measures at HSPNs with and without similarity cutoff. This observation supports that changes in the HSPN topology by removing edges (similarity relations) not only produce sparser networks with an increased number of clusters, but also lead to a variation in peptides’ centrality measures and therefore the networks’ topological distribution. However, a small set made up of likely the most relevant peptides was identified for four and three centrality measures in both HSPNs. These are the cases of starPep_00048, starPep_03668, starPep_10922 and starPep_00000. These peptides seem to be very important within the ABF chemical space. The starPep_00048 is a human defensin derivative (HNP1) of 30 amino acid length, with several reported interesting bioactivities (antiviral, anti-Gram+/−, antifungal, anticancer, enzymatic inhibitor) alongside its antibiofilm action, which is probably exerted by its ability to disrupt membranes and interfere in biological processes involved in interspecies interaction [50,51]. The starPep_03668 and starPep_10922 peptides are synthetic constructions of 35 and 12 aa length, respectively. The starPep_03668 was designed as pathogen-selective peptide, based on the fusion of a species-specific targeting peptide domain with a wide-spectrum antimicrobial peptide domain. Thus, it showed activity against several communities of *Streptococcus* species [52]. By contrast, starPep_10922 was designed as a D-enantiomeric peptide aiming to resist proteases’ degradation and also to prevent the accumulation of (p)ppGpp, which is a key messenger for biofilm formation. StarPep_10922 was able to prevent biofilm formation from *P. aeruginosa* as well as to disperse and eradicate the bacteria in the resulting mature biofilm [53]. So far, these three central peptides have not been reported as toxic to mammalian cells, and are lead peptides for developing ABFP drugs. The peptide starPep_00000, a 26 aa length ABFP that was derived from melittin (bee venom), appears to be a promising candidate, since several pharmacological activities have been assigned to it quite apart from its antibiofilm activity; it has also been extensively evaluated against many targets [54,55]. However, starPep_00000 has also shown haemolytic activity and toxicity to eukaryotic cells, which may limit its therapeutic potential unless its toxicity can be relieved by optimization procedures [55].

The ABFP space is not only represented by the central peptides; there also exist peptides that are disconnected from the giant component of the network and which bear low values of node degree. These peptides are categorized as atypical because they share very low sequence similarities with the central or popular ones, which prevents the estimation of their properties. However, they are remote members of the ABFP family and they also account for the antibiofilm chemical space. In this sense, the HSPN with similarity cutoff is very useful for uncovering atypical peptides. The cutoff of AF similarity at 0.65 was the optimal for retaining a reasonable trade-off between the number of communities and singletons; where the simplest community is considered to be when two nodes (peptides) are connected, and singletons are those that are not connected with any other in the network. Atypical peptides represent singular structures that may represent privilege scaffolds for designing antibiofilm agents.

Since the HSPN with no cutoff is fully connected, no isolated communities and singletons can be identified. Table 2 displays the atypical peptides from the HSPN at a similarity cutoff of 0.65; the 20 singletons where all centrality measures reached 0 value and two isolated communities made up of two peptides interconnected with node degree 1.

The atypical peptides could have been analyzed similarly to the central peptides, but as all singletons displayed 0 values for all centrality measures, the analysis was carried out by exploring the metadata associated with each peptide using the StarPep toolbox. Atypical peptides with no reported toxicity to mammalian cells are marked with an asterisk, while those that also have a diversity of desired functions contributing to the antibiotic/antibiofilm activity were highlighted in bold. This is the case for starPep_04044, which is a synthetic peptide of 13 aa length representing a singular structure that can successfully coat titanium surfaces and can also target Gram+ and Gram− bacterial strains in both their planktonic and biofilm forms, thereby allowing its utilization for preventing infection-related implant failures in dentistry and orthopaedics. It has been proven effective on *Pseudomonas aeruginosa*, *Streptococcus gordonii*, *Porphyromonas gingivalis*, *Staphylococcus aureus* and *Escherichia coli* [56,57].

#### 3.2.2. Metadata Analysis by Visual Mining

The METNs corresponding to the 174 ABFPs were constructed considering their source database and origin. Similar to the previous visualization, nodes were displayed by color and size. Red nodes represent source databases and origin, respectively, while the ABFPs are in gray (Figure 5A,B). Node size was scaled according to their betweenness centrality values, which is based on the shortest path between two nodes. Thus, this type of centrality is the number of shortest paths that pass through a target node, particularly on the red nodes representing “database” and “origin”.

As shown in Figure 5A, the 174 ABFPs registered in StarPepDB were mainly collected from BaAMP [27], the Structurally Annotated Therapeutic database (SATPdb) [58] and Database of Antimicrobial Activity and Structure of Peptides (DBAASP) [59], which are represented by larger nodes with the highest connection with the ABFPs. While BaAMP and DBAASP entries were carefully collected from the literature, the SATPdb was built from 22 peptide databases that included BaAMP [27] and others similarly dedicated to specific activities such as AVPdb (antiviral) [60], ParaPep (anti parasitic) [61], Hemolytik (hemolytic) [62], CancerPPD (anticancer) [63], etc. It was also built from generic AMP databases such as DAMPD [64], APD [25], CAMP [65], LAMP [66], DRAMP [67], etc. Such databases that integrated SATPdb were represented as smaller red nodes, sharing fewer edges with the ABFPs. 

Most of the ABFPs come from synthetic constructs and are represented by the largest node or hub in the network (Figure 5B). However, other natural sources have also provided ABFP scaffolds for further modification/optimization. Among the most contributing taxonomic groups are the Bacteria, Homininae, Similiformes, Pan and Bos (Bos Taurus). Therefore, this information can guide the discovery and design of antibiofilm agents from peptides. Particularly, it can inform us of the most relevant ABFPs databases and the main sources/origins of promising ABFP scaffolds.

### 3.3. Representing the ABFPs with a Reduced Subset

#### 3.3.1. The Selection of the Best Representative Subset

The original space of 174 ABFPs, illustrated by the HSPN model at the optimal cutoff of 0.65 (Figure 4B), can be further simplified by applying the scaffold extraction algorithm implemented in the StarPep toolbox. This algorithm allows the topological simplification of the network by removing nodes with equal or similar values of centrality measures but retaining those that still share a local similarity below a certain cutoff. As the reduction of ABFPs intends to keep the HSPN topology and properties, it is applied to all type of nodes, from the most central to the atypical ones (singletons). The reduction was performed by ranking the harmonic and hub-bridge centrality values of the nodes and by applying similarity cutoffs from relaxed criteria (retaining all peptides sharing < 0.9) to more restrictive similarities (<0.45 of sequence identity). This step produces several subsets for each centrality metric (Table 3 and Appendix A). 

As mentioned before, the best subset produced by the HSPN model and representing the original space should be composed of a minimum number of ABPFs with a coverage of the original space of <50%. However, the coverage is not the unique criterion for selecting a representative subset of the original space; the distribution of the HSPN model in the bi-dimensional (2D) space should also be considered. An effective subset should have topological representativeness over all the network, representing connected communities, isolated communities and even singletons. In this sense, a subset extracted under the criterion of only one centrality measure might not fulfil the expected 2D coverage. Thus, we decided to fuse the information of the HC and HB centralities, since they rank the network nodes according to their position using different definitions. Thus, the union of the subsets 6, 7 and 8, which are highlighted in bold in Table 3, was evaluated. Subsets 6, 7 and 8 display a coverage of <50% with a low of number of ABFPs, but are promising for their union (HC ∪ HB) at the same cutoff value. It is notable that the union between two subsets includes the common peptides (intersection) and the singular peptides from each subset. The union of subsets 6, 7 and 8 resulted in fasta files (Appendix A) containing 85, 66 and 52 ABFPs, representing 49%, 38% and 30% of the HSPN model, respectively. The Appendix A also contains the 221 ABFPs registered in StarPepDB, and the 174 used to generate the HSPNs. 

Finally, the subsets 6, 7 and 8 from each centrality metric as well as their resulting fusion (HC ∪ HB) were overlapped on the HSPN model to evaluate their 2D coverage/spatial distribution (Appendix A). In the three subsets, the union of the centrality measures (HC ∪ HB) at the evaluated similarity cutoffs displayed a better 2D spatial distribution on the HSPN model. Particularly, the HC ∪ HB from subset 7 showed the best trade-off, considering the lowest number of peptides with the best 2D coverage (Appendix A). Figure 6 summarizes the overlapping of the HC ∪ HB from subset 6, 7 and 8 with the HSPN model. The overlapped subsets are represented by small black nodes over the colored nodes that represent the communities in the HSPN model. 

#### 3.3.2. Visualizing/Analyzing the Best Representative Subset with HSPNs

The main goal of extracting a reduced subset representing the complex networks is to enable the retrieval of useful information from visual inspection of the networks. With a reduced number of nodes and edges, complex networks become more legible for the human eyes, and the representativeness of the subset is also useful for multi-reference similarity searches against unlabeled peptides. Figure 7 shows the HSPN constructed from the reduced space, which is made up of 66 ABFPs resulting from the HC ∪ HB of the subset 7 (cutoff 0.40). As can be seen in Figure 7A,B, most of the main ABFPs identified in the HSPN model were transferred to the reduced space; such is the case for starPep_00000, starPep_00042, starPep_00048, starPep_03668, starPep_05561 and starPep_00004, which additionally were among the ten most relevant ABFPs, as ranked by the HC and HB centralities (Table 1). Its noteworthy that other ABFPs not identified among the top-ranked in the HSPN model were brought up significantly when constructing the HSPN with 66 representatives, e.g., starPep_00522 and starPep_00514 (Figure 7A,B and Appendix A). 

On the other hand, 5 out of the 20 singletons appeared in the representative subset. They are starPep_00002, starPep_04044, starPep_05305, starPep_09934, and starPep_10637, with all being reported as non-toxic except starPep_00002. It was not by chance that the privileged scaffold of starPep_04044 was also selected, which indicates that the scaffold extraction algorithm implemented in the StarPep toolbox works.

The subset of 66 representative ABFPs extracted from the HSPN model is described in Appendix A, which includes their IDs, sequences, lengths, the cluster to which they belonged in the HSPN model, and their centralities measures both in the HSPN model and in the newly constructed HSPN (Figure 7). This non-redundant subset of 66 peptides represents the chemical space of the ABFPs, and it is advisable use it as a reference to map new ABFP sequences as well as in multi-reference similarity searches against unlabeled peptide datasets and for design purposes. Other network-based analyses of the metadata associated with these 66 ABFPs will provide a different insight for the selection of privileged scaffolds for the development of next-generation antimicrobials.

#### 3.3.3. Visualizing Mining of the METNs

The reduced subset is also useful for unravelling information from METNs, which are even more complex than similarity networks because they include other layers containing additional nodes with associated peptide metadata. Consequently, counting on a representative subset of the original space aids in analyzing the huge amount of information associated with ABFPs. As previously mentioned, microbial biofilms are responsible for most chronic and medical device-related infections, as well as for microbial resistance to several antibiotic classes; thus, the identification of promising ABFP scaffolds is an urgent task.

As an illustrative example of the METNs’ contribution to the identification of promising ABFPs scaffolds, six relevant ABFPs according to the HC and HB centralities were selected from the representative subset (Figure 7) for visual analysis of their metadata. This time, key metadata for the development/design of ABF agents from ABFPs were chosen, e.g., other associated activities of the ABFPs and the targets that they have been evaluated on (Figure 8). The study cases were starPep_00000 (blue), starPep_00193 (yellow), starPep_00004 (green), starPep_00025 (pink), starPep_00514 (black), and starPep_00522 (cyan).

Figure 8A is organized in such a way that the desired activities were placed outside at the left part of the METN, and the undesired ones at the right. The six candidates under study, in addition to antibiofilm activity also have antifungal and the antibacterial properties, specifically against Gram-positive and Gram-negative strains, which are very convenient for developing next-generation antimicrobial agents able to target both planktonic and biofilm microbial forms. However, four of them were reported as hemolytic, and five as “toxic to mammals”. The peptide starPep_00522 (in cyan) is the only one that was neither reported as “hemolytic” nor “toxic to mammals”, and therefore its peptidic scaffold can be used for developing antibiofilm agents for clinical purposes. Additionally, Figure 8B also shows that starPep_00522 has been evaluated in *Escherichia. coli*, *Pseudomonas aeruginosa*, *Candida albicans*, and *Cryptococcus neoformans*, of which the two first targets are classified as ESKAPEE pathogens, considered the most threatening antimicrobial-resistant microbes [68]. This visual analysis of the METNs can be extended to all 66 ABFPs of the representative set. 

### 3.4. External Representative ABFPs on the Representative Antibiofilm HSPN

In a recent work, Li at al. arrived at 14 representative ABFPs out of a total of 51 peptides with a reported antibiofilm activity either through inhibition of the biofilm formation or through the eradication of pre-formed biofilms. The selection was based on the identification of 14 ABFP classes according to their mechanisms of action. They also evaluated them against the biofilm and planktonic forms of Gram-positive bacterium Streptococcus mutans, the Gram-negative bacterium Pseudomonas aeruginosa, and the fungus Candida albicans. Those ABFPs with minimal biofilm inhibitory concentrations (MBICs) that are lower than their minimal inhibitory concentration (MICs) represented promising candidates against biofilm-related infections. Table 4 shows the 14 representative candidates categorized by their antibiofilm mechanisms of action (information taken from Appendix A published in [69]). 

This set made up of 14 ABFP classes representing different mechanisms of action for carrying out antibiofilm activity was mapped on the structural/chemical space of the 66 ABFPs, drawn by the HSPN model (Figure 9A). As the subset of 66 ABFPs showed the best representativeness of the antibiofilm chemical space, it was used along with the most suitable HSPN projection for overlapping purposes. 

The HSPN that plots nodes coordinates from their two most relevant principal components, estimated from a non-redundant set of AF descriptors, is the most real approach to display the peptide’s location in the network, allowing a more accurate visual inspection of the similarity and distribution of the 14 representative ABFPs on the reported chemical space (Figure 9A).

Figure 9A shows the node and names corresponding to 14 ABFPs in black color, while the other 66 from StarPepDB were labeled according to the colors assigned to each one of the five network communities. As can be observed in Figure 9A, all 14 ABFPs were framed within the antibiofilm HSPN. In fact, the Indolicidin, Protegrin-1 and HBD-3 overlapped perfectly with starPep_00002, starPep_00020 and starPep_00116, respectively. The 14 mechanisms of action classes were distributed among all the five HSPN communities, which may indicate a connection between structural patterns (motifs) found within network communities with the antibiofilm mode of action. In order to illustrate this fact, six ABFPs that showed antibiofilm activity against both bacteria and fungi (pleurocidin, Pac-525, protegrin-1, TetraF2W-RR, WLBU2, and melittin) overlapped perfectly or almost perfectly over different communities. Pleurocidin, Pac-525, protegrin-1 and melittin are evidently overlapped on the communities colored in blue, green, pink and light purple. WLBU2 was also placed in the green community as the Pac-525, probably because both act using similar mechanisms of action involving the interaction with the lipopolysaccharides to destroy or penetrate the bacterial membrane. Although TetraF2W-RR is within the antibiofilm space represented by the HSPN, it did not overlap on any specific community. However, it was placed between green and pink communities containing members such as Pac-525, WLBU2, Indolicidin and protegrin-1, whose mode of action is closely related to that reported for TetraF2W-RR (bacterial membrane disruption). These four ABFPs are arginine ®-rich peptides containing repeated units of R, allowing interaction with negatively-charged bacterial membranes, the formation of transmembrane pores, and cell penetration [70].

Figure 9B complements the information extracted from Figure 8A. It shows those representative ABFPs sharing AF similarities > 0.60 with some of the 66 ABFPs extracted from StarPepDB. Black nodes represent the 9 out of the 14 representative ABFPs that fulfill this condition, while the black edges display the similarity relationships between black nodes (origin) and colored nodes (target). Target nodes labeled as starPep_XXXXX retained the same color identifying them at the network communities in Figure 9A. Figure 9B confirms that Indolicidin, Protegrin-1 and HBD-3 share the max. similarity (1.0) with those ABFPs that overlapped with starPep_00002, starPep_00020 and starPep_00116 in Figure 9A. The location of pleurocidin and protegrin-1 was also supported by their highest similarities (0.65 and 1.0), with members of the communities blue (starPep_00496) and pink (starPep_00020), respectively. However, as the pleurocidin shows multiple actions such as membrane disturbance and permeabilization, binding to bacterial DNA and interference with several cellular functions, it also displays similarities with members (starPep_00193, starPep_00051) from two other communities (Figure 9B).

On the other hand, Figure 9B also served to correct the location of melittin that is actually overlapped over the orange community, showing 0.95 similarity to starPep_00000 (a melittin derivative), despite looking over the light purple community in Figure 9A. Although the 14 ABFPs could be mapped within the HSPN, Figure 9B confirmed the certain singularity of Pac-525, peptide 1037, TetraF2W-RR, P1 and WLBU2 within the representative ABFP space, not sharing AF similarities higher than 60%. Such singularity was also confirmed by evaluating the pairwise identity of these last 5 ABFPs against the 66 representative ones (Figure 10B). The 9 ABFPs that were clearly mapped at AF similarities > 0.60 were also compared by pairwise global alignments (Figure 10A).

The lower part of Figure 10A, framed by the white line, displays five red dots that correspond to those ABFPs (Indolicidin, Protegrin-1, HBD-3, Melittin and Nisin) sharing network edges weighted with AF similarities higher than 0.90; the edge weighted with 0.69 is likely represented by the yellow dot, and the remaining edges slightly above 0.60 are depicted in cyan colors. Figure 10B confirmed that the similarities shared by the all five unmapped ABFPs were actually below 0.60. All dots were mostly colored in blue, and a few in cyan may represent values close to but below 0.60. 

It is important to note that as the AF and AB similarities are defined under different methodological frameworks, they may characterize the same pairwise relation with different values, despite being correlated. The Appendix A shows the pairwise identity values of nine and five ABFPs from the 14 mode-of-action classes against the 66 ABFPs representing the antibiofilm chemical space. 

### 3.5. Motif Discovery Assisted by Complex Networks

The identification of motifs accounting for the antibiofilm activity can be assisted by the exploration of ABFP similarity networks looking for sequence patterns within network communities. Although the HSPN representing the ABFP chemical space was built using AF distance metrics (Euclidean) and the network communities are estimated considering parameters from the nodes and edges properties [40], such clusters should contain peptides sharing similar features. Thus, the communities from the HSPN model of the 174 ABFPs resulted in the source for the motif discovery. The sequence diversity in each community was evaluated by global alignments. Appendix A displays the heatmaps that mirror the pairwise sequence identities for communities containing more than two peptides that correspond to the clusters 4, 7, 9, 11, 14, 15, 17, and 22, including clustered singletons (Appendix A). Appendix A evidenced a high sequence diversity within all communities. Consequently, iterative alignment algorithms such as MAFFT and MUSCLE were applied to deal with the high sequence diversity. The multiple sequence alignments (MSAs) were visualized with the Jalview, which allowed the estimation of their corresponding consensus sequences and Seq2Logos. The consensus sequences from the MSAs were also estimated by the EMBOSS Cons. The full exploration of the MSAs considering their corresponding consensus sequence and Seq2Logos allowed the identification of conserved regions considered ABF motifs. The strategy carried out for the identification of the motifs in the MSAs and performed on cluster 4 of the HSPN is illustrated in Figure 11, while the strategy for all communities/clusters is displayed in Appendix A.

Table 5 listed ABFP motifs identified by each MSA method at each network community or cluster. The consensus estimated by the EMBOSS Cons was the preferred template for motif identification, because it gives a more legible output. High-scoring amino acids/positions are represented by capital letters, and lower-scoring but positive residues by lower-case letters; non-consensus positions are denoted by x (Table 5). 

As part of the motif discovery process, the AB search was complemented by evaluating an AF approach. Based on its high performance and versatility in identifying motifs in OMICs data, the STREME algorithm was applied to find unaligned patterns ranging from 3–5 aa length in each network community [46]. STREME computes a score for the detected motifs meeting the threshold of statistical significance (*p*-value < 0.05), set also as a stopping search criterion. Table 6 displays the discriminating motifs against control sequences in each ABFP cluster/community. Motifs appearing in more than 20% of the query peptides are listed according to their statistical significance (score).

Motifs highlighted in bold in Table 5 and Table 6 are closely related or included in each other. This means that both the MSA algorithms and STREME showed some degree of agreement in their motif detection. However, both approaches also identify singular motifs, which are not highlighted. This fact demonstrates that the application of both AB and AF approaches was the right choice for a full motif exploration. Given that both methods identified a relatively high number of motifs between 33–35, enrichment analyses were further performed in order to filter the discovery motifs shown in Table 5 and Table 6. 

Motif enrichment analyses are used to determine if a group of sequences contains a statistically significant number of matches to a given motif. In this sense, we used the SEA algorithm [48] to select which motifs from both tables were significantly enriched in two sets of ABFPs. The first set was that reported by Li et al. consisting of 14 representative ABFPs of antibiofilm modes of action [69], and the second one encompassed 192 non-redundant ABFPs, extracted from the 214 ABFPs registered in BaAMP database [27] (Appendix A). Eight members from the representative subset were included among the 192 ABFPs, but all of them have shown antibiofilm activity at different levels. As a screening criterion, ABFPs enriched in both the representative and the extended dataset were selected. Table 7 lists the ABFP motifs discovered by the AB and AF approaches in the network communities.

Our motif search approach assisted by complex networks produced results that are not so far from the few findings reported in the literature. Recently, Anastasiu et al. found that the following motifs “RIRV,” “RIVQRIK,” and “IGKEFKR” appeared with more frequency in 242 ABFPs collected from APD and BaAMP databases with respect to a curated negative set [31], when using the “MERCI” software [72]. In this sense, we agree with them in the detection of the “RIRV” which was fully integrated in the RIRVR motif detected in cluster 14 by the MAFFT algorithm, and also enriched in the BaAMP dataset. Although “RIVQRIK” and “IGKEFKR” were not detected as such, we could identify in clusters 15 and among the singletons by the MSA methods, the “RIV” and “FIK” patterns, which are part of them. These two last three-amino acid motifs were also enriched in the extended dataset.

In a previous report, the authors also found that the dipeptides “IR/RI”, “WR/RW”, and “KK” were the most common among the selected ABFPs [31]. Certainly, these dipeptides are present in the motifs discovered with the intervention of complex networks, at a relatively high frequency. For example, the “IR/RI”, “WR/RW” and “KK” dipeptides appear in 9/7, 7/8 and 12 of the total motifs, respectively. In addition to them, the “RR” and “KL” dipeptides also displayed a similar representation among the motifs. 

In a recent report, arginine-rich motifs of the antibiofilm activity of peptides designed for sequestering the nucleotide second messenger c-di-GMP, involved in the formation of *P. aeruginosa* and *K. pneumoniae* biofilms, were revisited [73]. The key role of the DRR and [RK] RxxD motifs in these sequestering peptides (SP) to specifically bind to c-di-GMP was demonstrated by nuclear magnetic resonance (NMR)-based experiments [18]. These motifs associated with CSPs are difficult to discover using bioinformatics methods, since their sourcing peptides are probably still not registered or underrepresented in databases. However, the peptide R4F4 (RRRRFFFF), with a proven antibiofilm activity on *P. aeruginosa* through c-di-GMP sequestration [73,74], bears a more frequent motif (RRRR) among the arginine-rich ABFPs. In fact, “RRRR” was detected in our complex network-assisted motif search (Table 5 and Table 6). 

On the other hand, the role of the WWW motif in disrupting preformed biofilms of methicillin-resistant *Staphylococcus aureus* was elucidated by NMR and arginine scan experiments in 2017 [75]; the WWW motif hardly appears in ABFP databases, being only represented by the designed peptide TetraF2W-RR [76]. 

As both the motif discovery and the distinction of ABFP hallmarks are highly influenced by peptide databases’ composition and by the searching algorithm, here, we provide new ABFP motifs discovered from combining network science with AB- and AF-based computational tools for motif detection. Unlike the study reported in [31], our approach hardly depends on the selection of a negative set to retrieve relevant information from the reported ABFPs; this may represent an advantage, since final outcomes/conclusions could be biased by the selection of a negative set [77]. The motifs listed in 5, 6 and especially in Table 7 are useful for the “in silico” generation of peptide libraries addressing antibiofilm and antimicrobial activities, as well as for the optimization of antibiofilm candidates for clinical purposes. Finally, predicted motifs that actually account for/improve antibiofilm activity could be used as motif-based descriptors for developing machine-learning models to screen peptide libraries and peptidomes as part of the discovery process.

## 4. Conclusions

Half-space proximal networks were successfully introduced to project the pairwise alignment-free similarities of the reported ABFPs. Particularly, an HSPN model was obtained by applying an optimal similarity cutoff of 0.65, which allowed an effective delineation of network communities with the respective identification of the most central (relevant) and atypical peptides among the ABFPs. From the topology of the HSPN model, a reduced subset of 66 ABFPs, resulting from the union of the harmonic and hub-bridge centralities, was extracted by the scaffold extraction algorithm from the StarPep toolbox. As these 66 peptides, made up of both the most relevant and atypical ABFPs, were not only selected by the centrality criteria but also considering their topological distribution and coverage on the original space, they can be considered representatives of the ABFP chemical space.

Alongside this previously mentioned procedure, the metadata associated with both the most central and atypical peptides were analyzed by the visual mining of complex networks, integrating additional relevant properties for the antibiofilm action and for the discovery/design of next-generation antimicrobial agents able to combat MDR infections. On the other hand, the proposed network-assisted motif discovery allowed the identification of ABFP motifs by AB and AF approaches within the communities of the HSPN model.

In short, the network-based identification of the most central to atypical ABFP scaffolds bearing promising antimicrobial activities on MDR targets was mostly transferred to the representative subset of 66 ABFPs. Therefore, this subset together with the discovered ABFP motifs is recommended for use in the mapping and design of new ABFPs as privileged scaffolds for the development of next-generation antimicrobials.

This is the first work that illustrates how complex networks can be integrated in the discovery pipeline of more effective antimicrobial agents from the existing data. As an emerging methodology in the field, it still has many uncovered potentialities and will need the support of existing methodologies, as stated here.

## Figures and Tables

**Figure 1 antibiotics-12-00747-f001:**
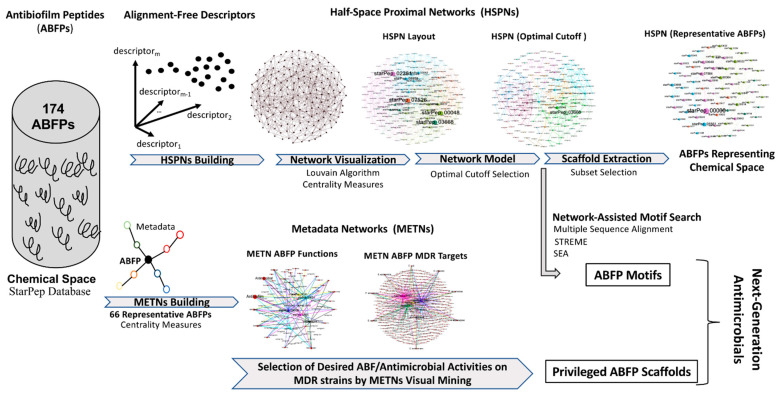
Overview of the proposed workflow integrating complex networks for next-generation antimicrobials’ development from ABFPs.

**Figure 2 antibiotics-12-00747-f002:**
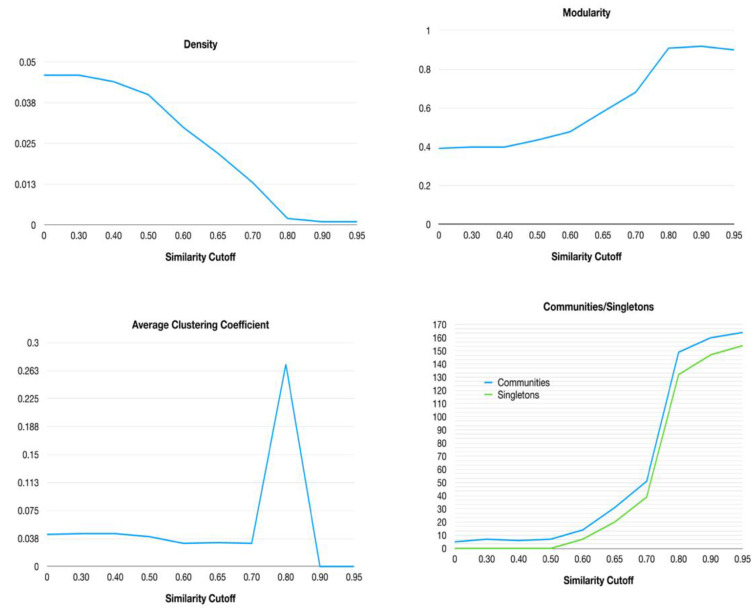
Behavior of HSPNs parameters representing the 174 ABFPs when the similarity cutoff varies from 0 to 0.95.

**Figure 3 antibiotics-12-00747-f003:**
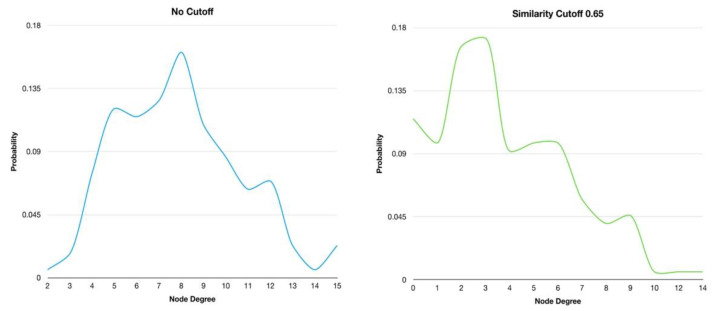
Degree distribution of the HSPNs corresponding to the 174 ABFPs with no similarity cutoff and after applying a cutoff of 0.65.

**Figure 4 antibiotics-12-00747-f004:**
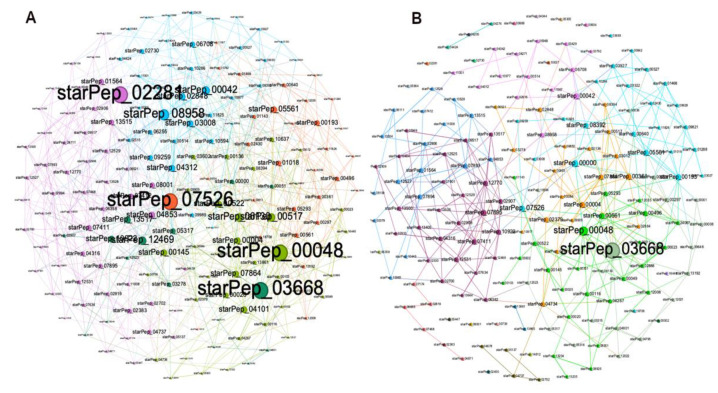
(**A**) HSPNs’ visualization using the Fruchterman–Reingold layout, without similarity cutoff and (**B**) with a similarity cutoff of 0.65. Peptides communities are represented by different colors, while the nodes’ size was scaled according to node degree.

**Figure 5 antibiotics-12-00747-f005:**
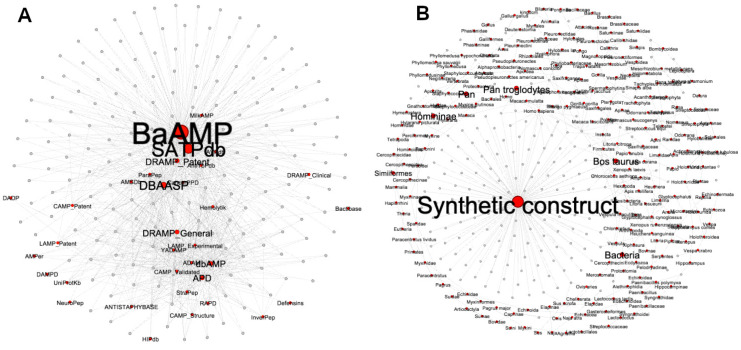
METNs’ visualization using the Fruchterman–Reingold layout. Red nodes represent the metadata, while the ABFPs are in gray color. (**A**) METN built with the source database information. (**B**) METN built with the origin information.

**Figure 6 antibiotics-12-00747-f006:**
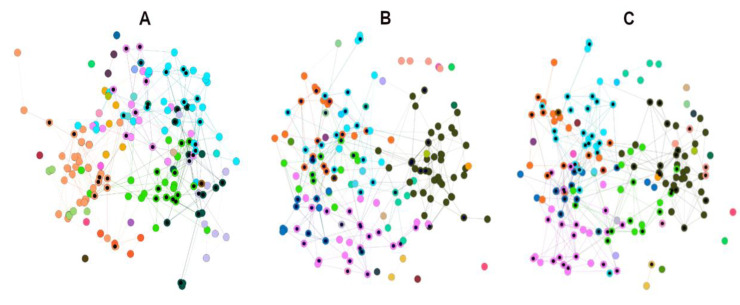
Overlapping subsets 6, 7 and 8 resulting from the union of the harmonic and hub-bridge centralities at 0.35, 0.40 and 0.45 over the HSPN model. (**A**) 52 ABFPs from the HC ∪ HB of the subset 8 (cutoff 0.35) over the total 174 ABFPs; (**B**) 66 ABFPs from the HC ∪ HB of the subset 7 (cutoff 0.40) over the total 174 ABFPs; (**C**) 85 ABFPs from the HC ∪ HB of the subset 6 (cutoff 0.45) over the total 174 ABFPs.

**Figure 7 antibiotics-12-00747-f007:**
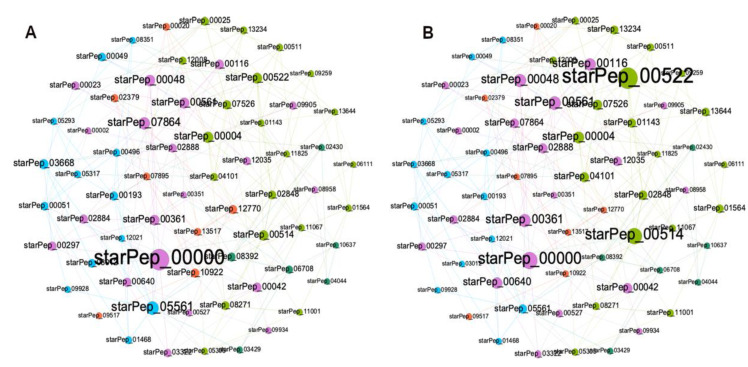
HSPNs visualization of the representative subset integrated by 66 ABFPs from the HC ∪ HB of the subset 7 (cutoff 0.40) using the Fruchterman-Reingold layout. Peptides communities are represented by different colors while the nodes size was scaled according to harmonic (**A**) and hub-bridge centrality (**B**) measures.

**Figure 8 antibiotics-12-00747-f008:**
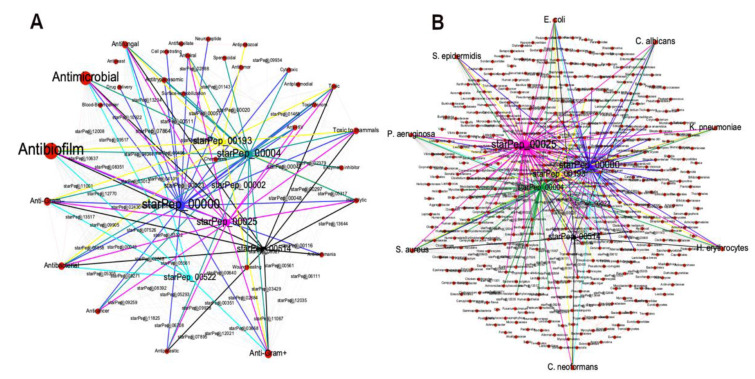
METNs visualization of the representative subset integrated by 66 ABFPs from the HC ∪ HB of the subset 7 (cutoff 0.40) using the Fruchterman–Reingold layout. (**A**) Red nodes represent the bioactivities, while the ABFPs under study, starPep_00000, starPep_00193, starPep_00004, starPep_00025, starPep_00514, and starPep_00522, are in blue, yellow, green, pink black and cyan colors, respectively. (**B**) Red nodes represent the targets on which the ABFPs have been evaluated. The ABFPs under study are highlighted with the color scheme used in A.

**Figure 9 antibiotics-12-00747-f009:**
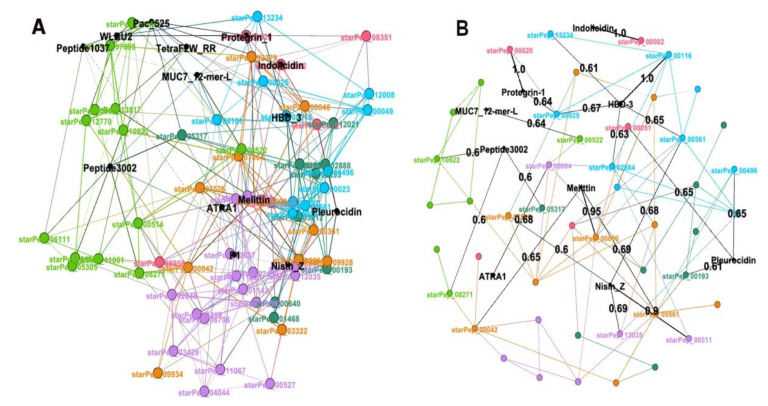
Visual mining of the 14 ABFPs corresponding to 14 classes of mode of actions on the chemical space of ABFPs. (**A**) Overlapping the 14 representative ABFPs on the set integrated by 66 ABFPs using HSPN. Peptide communities are represented by different colors, while the 14 ABFPs nodes/labels are highlighted in black color. (**B**) Mapping of the representative ABFPs with similarities higher than 0.60 within the antibiofilm HSPN using the Fruchterman–Reingold layout.

**Figure 10 antibiotics-12-00747-f010:**
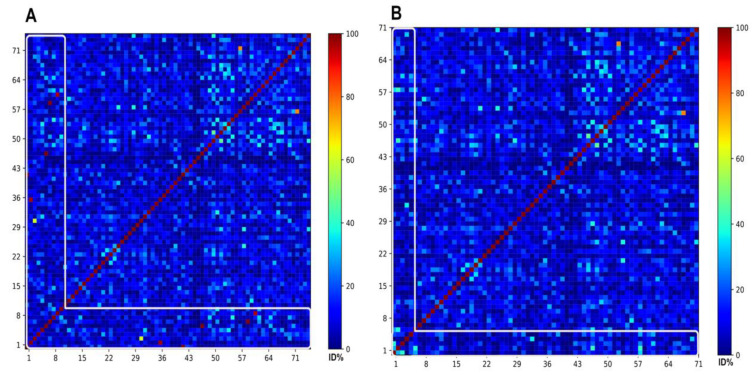
Heat maps corresponding to the pairwise identities from the comparison of the 14 ABFPs representing the mechanisms of action classes against the 66 representatives of the antibiofilm chemical space. (**A**) the 9 ABFPs that could be mapped on the representative HSPN at AF similarity >0.60 occupy the from 1 to 9 position in the heatmap. (**B**) the 5 that did not map at AF >0.60 were placed from 1 to 5 position. The target zone of the heat maps is framed by a white line. All-*vs*-all global alignments and heat maps visualization were conducted using the SeqDivA software reported in [71].

**Figure 11 antibiotics-12-00747-f011:**
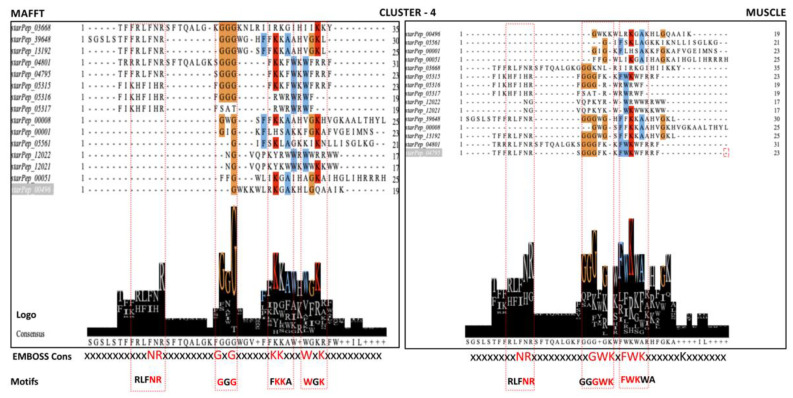
Motif detection by the multiple sequence alignment (MSA) algorithms MAFFT and MUSCLE on the network cluster 4. The MSAs are visualized with the Jalview program, which also estimates a Seq2Logo and the consensus from the alignment positions. Another consensus sequence that served as a guide for motif location was estimated by the EMBOSS Con.

**Table 1 antibiotics-12-00747-t001:** Common and singular peptides from the ten top-ranked ABFPs by four, three, two and one centrality measure (node degree, harmonic, betweenness and hub-bridge centralities). The cluster number for each peptide at each HSPN is displayed.

HSPN—No Cutoff			
Centrality Measure	Total	Peptide Name	Cluster
Node Degree Harmonic Betweenness Hub-Bridge	1	starPep_03668	(1)
Node Degree Betweenness Hub-Bridge	1	starPep_00048	(1)
Harmonic Hub-Bridge	2	starPep_00000 starPep_10922	(1) (3)
Node Degree Betweenness	7	starPep_12469 starPep_07526 starPep_00145 starPep_06130 starPep_00042 starPep_08958 starPep_02281	(3) (2)(2) (3) (0) (2) (2)
Node Degree	1	starPep_00517	(0)
Harmonic	7	starPep_07864 starPep_07895 starPep_02907 starPep_12770 starPep_12531 starPep_13517 starPep_07893	(0) (3)(3) (3)(3) (3) (2)
Betweenness	1	starPep_08001	(3)
Hub-Bridge	6	starPep_00496 starPep_00561 starPep_00361 starPep_13515 starPep_00193 starPep_05561	(1) (1) (1) (2) (1) (1)
**HSPN—Cutoff 0.65**			
Node Degree Harmonic Betweenness Hub-Bridge	1	starPep_00048	(9)
Node Degree Betweenness Hub-Bridge	1	starPep_00042	(11)
Harmonic Betweenness Hub-Bridge	1	starPep_10922	(15)
Node Degree Harmonic Betweenness	2	starPep_00000 starPep_03668	(15) (4)
Node Degree Harmonic	1	starPep_00361	(9)
Node Degree Betweenness	1	starPep_07526	(17)
Node Degree Hub-Bridge	2	starPep_00004 starPep_00193	(11) (9)
Harmonic Betweenness	1	starPep_02379	(17)
Harmonic Hub-Bridge	3	starPep_07895 starPep_02907 starPep_12531	(15) (15) (15)
Node Degree	2	starPep_00561 starPep_05561	(9) (4)
Harmonic	1	starPep_07893	(14)
Betweenness	3	starPep_12530 starPep_04734 starPep_08958	(15) (17) (15)
Hub-Bridge	2	starPep_12770 starPep_02908	(15) (15)

**Table 2 antibiotics-12-00747-t002:** Atypical peptides identified by the HSPN at a similarity cutoff of 0.65. Particularly, the singletons and isolated communities made up of two peptides are detailed.

HSPN—Cutoff 0.65			
Atypical Peptides	Total	Peptide Name	Cluster
Singletons	20	starPep_00002 starPep_00739 starPep_02281 starPep_02383 starPep_02400 starPep_02730 starPep_03693 **starPep_04044 *** starPep_05305 * starPep_05447 * starPep_05964 starPep_06255 starPep_06358 * starPep_08001 starPep_09934 * starPep_09989 * starPep_10637 * starPep_14812 * starPep_16445 * starPep_18706 *	(10) (18) (20) (25) (0) (5) (19) (13) (8) (27) (26) (21) (23) (24) (16) (29) (1) (12) (2) (3)
Isolated Community	2	starPep_04274–starPep_04424 starPep_13860 *–starPep_13861 *	(6) (28)

* Atypical ABFPs with no toxicity reported. Those highlighted in bold show a variety of activities other than antibiofilm.

**Table 3 antibiotics-12-00747-t003:** ABFP subsets extracted from the HSPN model made up of 174 nodes and 325 edges using the scaffold extraction algorithm from the StarPep toolbox. The harmonic and hub-bridge centralities were applied for the reduction step at different similarity cutoff values.

Harmonic (HC)					Hub-Bridge (HB)		
Subsets	Cutoff	Edges	Nodes	Coverage ^1^ %	Edges	Nodes	Coverage ^1^ %
1	0.90	227	154	89	276	154	89
2	0.80	230	138	79	235	137	79
3	0.70	199	122	70	201	125	72
4	0.60	167	103	59	162	104	60
5	0.50	128	80	46	112	80	46
**6**	**0.45**	**115**	**74**	**43**	**88**	**68**	**39**
**7**	**0.40**	**74**	**54**	**31**	**63**	**51**	**29**
**8**	**0.35**	**62**	**45**	**26**	**44**	**40**	**23**
9	0.30	40	32	22	35	34	20

^1^ Coverage is the percentage representing each subset of the 174 ABFPs.

**Table 4 antibiotics-12-00747-t004:** Representative ABFPs categorized in 14 classes according to the mechanisms of action elucidated for their antibiofilm activity. The information displayed in this table was gathered from Appendix A published in ref. [69].

Class-Peptides	Action Mode	Sequence
Class1- HBD-3	Influences *icaAD* and *icaR* genes’ transcription levels	GIINTLQKYYCRVRGGRCAVLSCLPKEEQIGKCSTRGRKCCRRKK
Class2- Nisin Z	Decreases adhesion, kills bacteria, reduces biofilm formation	ITSISLCTPGCKTGALM GCNMKTATCNCSIHVSK
Class3- MUC7 12-mer-L	Attracted to bacterial surfaces by the electrostatic bonding	RKSYKCLHKRCR
Class4- ATRA1	Promotes biofilm dispersal	KRFKKFFKKLKNSVK KRFKKFFKKLKVIGVT FPF
Class5- Pleurocidin	Induces disturbance/permeabilization of the membranes and bindssss to bacterial DNA, causing interference with cellular functions	GWGSFFKKAAHVGK HVGKAALTHYL
Class6- Pac-525	Able to enter membranes and to affect the lipopolysaccharides of Gram-negative bacteria	KWRRWVRWI
Class7- peptide 1037	Decreases the attachment of bacterial cells, stimulates twitching motility, and influences two major quorum sensing systems	KRFRIRVRV
Class8- Indolicidin	Induces lipid removal and mixed indolicidin–lipid patches alongside membrane permeabilization	ILPWKWPWWPWRR
Class9- Protegrin-1 *	Forms amyloid fibers to associate with the bacterial membrane and produce transmembrane pores	RGGRLCYCRRRFCVCVGR
Class10- Peptide 3002	Blocking (p)ppGpp	ILVRWIRWRIQW
Class11-TetraF2W-RR *	Disrupts membranes to kill bacteria rapidly	WWWLRRIW
Class12-P1	Interferes with the proper secretion and/or intermolecular interaction of key extracellular polymers in the biofilm matrix	PARKARAATAATAATAATAATAAT
Class13-WLBU2 *	LPS-binding property interferes with bacterial attachment, destroying the bacterial membrane	RRWVRRVRRVWRRVVRVVRRWVRR
Class14-Melittin *	Inhibits the expression of biofilm-associated *bap* genes	GIGAVLKVLTTGLPALISWIKRKRQQ

* Promising antibiofilm candidates with lower MBICs than MICs.

**Table 5 antibiotics-12-00747-t005:** Motifs discovered by multiple sequence alignment (MSA) in each network cluster/community.

No	Motif	EMBOSS Cons.	Cluster	Cluster Size	MSA Method	Enrichment Ratio *
1	**RLFNR**	xxxNR	4	15	MAFFT/ MUSCLE	-
2	**GGG**	GxG			MAFFT	-
3	**GGG**WK	xxGWK			MUSCLE	(−)/(2.25)
4	**FKKA**	xKKx			MAFFT	(−)/(4.0)
5	F**WKWA**	FWK			MAFFT	(3.0)/(2.83)
6	WGK	WxK			MAFFT	(−)/(1.41)
7	**LLLLLKKK**	LLLLLKKK	6	2	Pair-Aligned	-
8	L**ISWIK**	lisxik	7	8	MAFFT	(−)/(2.71)
9	K**NKRK**	knkxk			MUSCLE	(3.0)/(2.22)
10	**KRKQ**	kxkQ			MAFFT	(3.0)/(−)
11	**R**[G**P**]**RVS**	rxRVS			MAFFT	(−)/(3.0)
12	RRPR	RRxR			MUSCLE	-
13	[G**R**]**GG**	xGG			MAFFT	(3.0)/(1.90)
14	GGRRRR	GGrrRR			MUSCLE	(−)/(2.0)
15	**RRRRR**	RRRRR			MAFFT/ MUSCLE	-
16	ISGI	Ixxx	9	23	MAFFT	-
17	**FKK**LL	xKKLL	11	27	MAFFT/ MUSCLE	(−)/(2.25)
18	**KKL**K				MAFFT	-
19	KKL				MUSCLE	-
20	LKK	LKK			MUSCLE	-
21	RIRVR	RIRVR	14	23	MAFFT	(−)/(1.58)
22	**RVIR**	xRVIR			MAFFT	(−)/(1.32)
23	V**RVIR**				MUSCLE	(−)/(2.83)
24	**R**[W**L**]R	RxR			MUSCLE	(1.57)/(−)
25	RIRRW	RIxRW	15	26	MAFFT/ MUSCLE	(−)/(4.0)
26	RI[VR]W					(−)/(1.67)
27	WVV	WVV			MAFFT	(−)/(1.44)
28	I[IR]R	IIxR			MUSCLE	-
29	W**LRK**	Wxxx	17	23	MAFFT	(−)/(2.50)
30	RWK	Rxx			MUSCLE	-
31	KKL	Kxx			MAFFT	-
32	KR[AK**L**]**RK**	KRxRK			MUSCLE	(6.0)/(3.0)
33	**WR**[**I**V]**R**	xRWR[IV]R	22	5	MAFFT/ MUSCLE	
32	FR**WRI**				MAFFT	(3.0)/(−)
33	RWRVR				MUSCLE	(−)/(1.63)
34	Y**APW**YN	YAPWYN	28	2	Pair-Aligned	-
35	[FI][**K**W]**RK**	iKrK	Singletons	20	MAFFT/ MUSCLE	(−)/(1.46)

* Enrichment ratio was evaluated on the 14 ABFPs categorized by action mode in ref. [69] (first value) and 192 non-redundant ABFPs extracted from 214 reported in BaAMP (second value).

**Table 6 antibiotics-12-00747-t006:** Motifs identified by STREME at each network cluster.

No	Motif	Cluster	Cluster Size	Matches in ABFPs	Matches in Control	Sites (%)	Score	Enrichment Ratio *
1	FKKA	4	15	7	0	46.7	3.3e-003	(−)/(3.33)
2	**GGG**R			7	0	46.7	3.3e-003	(−)/(2.11)
3	**W[KR]W**F			7	0	46.7	3.3e-003	(−)/(1.38)
4	FIH			6	0	40.0	8.4e-002	-
5	**RLFNR**			5	0	33.3	2.1e-003	-
6	**KKK**	6	2	2	0	100	1.7e-001	-
7	**LLLLL**			2	0	100	1.7e-001	-
8	**RGG**	7	8	8	0	100	7.8e-005	(3.0)/(1.56)
9	**ISWIK**			4	0	50	3.8e-002	(−)/(2.83)
10	**NKRK**Q			4	0	50	3.8e-002	-
11	**RPRVS**			3	0	37.5	1.0e-001	(−)/(3.71)
12	**RRRRR**			3	0	37.5	1.0e-001	-
13	SAC	9	23	16	1	69.6	3.3e-006	-
14	AKA			5	0	21.7	2.85e-002	-
15	CD[VI]			5	0	21.7	2.85e-002	-
16	IA[GVK]			5	0	21.7	2.85e-002	-
17	L**FKKL**	11	27	9	0	33.3	8.8e-004	(−)/(2.40)
18	KVLK			8	0	29.6	2.1e-003	(3.0)/(4.0)
19	KRFL			6	0	22.2	1.1e-002	(3.0)/(1.8)
20	V**RLR**I	14	23	12	0	52.2	3.5e-005	-
21	**RVIR**			10	0	43.5	2.8e-004	(−)/(1.32)
22	VWVI	15	26	14	3	53.8	1.3e-003	(3.0)/(3.0)
23	VIWRR			8	0	30.8	2.1e-003	(−)/(2.50)
24	**LRK**	17	23	9	0	39.1	7.4e-004	(3.0)/(1.27)
25	WRRK			6	0	26.1	1.1e-002	(−)/(1.67)
26	**WRIR**	22	5	5	1	100	2.4e-002	(−)/(3.25)
27	IRR			2	3	40.0	9.0e-001	(1.67)/(−)
28	**APW**TN	28	2	2	0	100	1.7e-001	(−)/(3.0)
29	K**KRK**	Singletons	20	2	0	10.0	2.3e-001	-
30	KKVVF			2	0	10.0	2.4e-001	-
31	LLKLL			2	0	10.0	2.4e-001	-
32	VKFK			2	0	10.0	2.4e-001	-
33	WRWR			2	0	10.0	2.4e-001	(−)/(1.64)

* Enrichment ratio was evaluated on the 14 ABFPs categorized by action mode in ref. [69]) (first value) and 192 non-redundant ABFPs extracted from 214 reported in BaAMP (second value).

**Table 7 antibiotics-12-00747-t007:** Summary of the discovered ABFP motifs per network community (cluster), enriched in two datasets.

No	Motif	Cluster	Method	Enrichment Ratio
1	FWKWA	4	MAFFT	(3.0)/(2.83)
2	KNKRK	7	MUSCLE	(3.0)/(2.22)
3	[GR]GG	7	MAFFT/STREME	(3.0)/(1.90)
4	KVLK	11	STREME	(3.0)/(4.0)
5	KRFL	11	STREME	(3.0)/(1.8)
6	VWVI	15	STREME	(3.0)/(3.0)
7	KR[AKL]RK	17	MUSCLE	(6.0)/(3.0)
8	LRK	17	STREME	(3.0)/(1.27)

## Data Availability

The StarPep toolbox software and the respective user manual are freely available online at: http://mobiosd-hub.com/starpep, accessed on 5 January 2023 (CAMD-BIR International network, Quito, Ecuador).

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
