# Peer review of "Complex Networks Analyses of Antibiofilm Peptides: An Emerging Tool for Next-Generation Antimicrobials’ Discovery"

_antibiotics, 2023, doi:10.3390/antibiotics12040747_

Round 1
Reviewer 1 Report
Dear authors
It seems that meaningful in silico analyses have been performed. The discovery of new drugs using computational studies is promising towards overcoming failure in antibiotic therapies.
The abbreviation "MDR" in the conclusion should be defined and described in the introduction.
Also, please state an statistics about morbidity and mortality due to biofilm-associated infections and MDR strains worldwide.
kind regards
Author Response
Reviewer’s comments (RC1): It seems that meaningful in silico analyses have been performed. The discovery of new drugs using computational studies is promising towards overcoming failure in antibiotic therapies.
Author’s Response (AR1): Many thanks for appreciating our work
The abbreviation "MDR" in the conclusion should be defined and described in the introduction.
AR2: It was corrected as your suggestion
RC2: Also, please state a statistic about morbidity and mortality due to biofilm-associated infections and MDR strains worldwide.
AR2: A new paragraph supported by the reference 11 was integrated into the Introduction section (lines 67-72).
“Globally, the prevalence of biofilm-associated multi-drug resistant (MDR) infections among hospitalized patients ranges from 17.9% to 100.0%, and are mostly caused by bacteria such as S. aureus, A. baumannii, K. pneumoniae, P. aeruginosa and E. coli that can also be found as MDR free-living strains. As consequence, 98 000 deaths are only registered annually in United States according to the US Centre for Disease Control and Prevention [11]”
Reviewer 2 Report
The manuscript by Agüero-Chapin et al. reports network-based identification of Antibiofilm Peptides (ABFP). Furthermore, using the sequence alignment approach, authors have discovered ABFP motifs that allowed to design and development of potential next-generation antimicrobials peptides.
This nice work provides an emerging tool complemented by classical bioinformatics for next-generation antimicrobial analyses to target planktonic and biofilm microbial forms. It substantially advances our understanding of antibiofilm peptides and opens new avenues to extract ABFPs via complex network mining methods. The paper is nicely planned, well-written, and well-illustrated. The methods are described in sufficient detail.
I have some minor comments for the authors.
Minor comments
The role of c-di-GMP in forming a biofilm is missing in the introduction.
The author should consider discussing the importance of c-di-GMP in biofilm formation. This will help readers to understand how the arginine-rich peptides intercept via sequestering second messenger c-di-GMP signaling in P. aeruginosa and K. pneumoniae biofilms.
L. 26. Change " threaten " to "threat. "
For clarity, I would suggest increasing the font size in Figure 4.
Author Response
Reviewer’s comments (RC1): The manuscript by Agüero-Chapin et al. reports network-based identification of Antibiofilm Peptides (ABFP). Furthermore, using the sequence alignment approach, authors have discovered ABFP motifs that allowed to design and development of potential next-generation antimicrobials peptides.
This nice work provides an emerging tool complemented by classical bioinformatics for next-generation antimicrobial analyses to target planktonic and biofilm microbial forms. It substantially advances our understanding of antibiofilm peptides and opens new avenues to extract ABFPs via complex network mining methods. The paper is nicely planned, well-written, and well-illustrated. The methods are described in sufficient detail.
I have some minor comments for the authors.
Author’s Response (AR1): Many thanks for your encouraging words on our work.
Minor comments
RC2: The role of c-di-GMP in forming a biofilm is missing in the introduction.
The author should consider discussing the importance of c-di-GMP in biofilm formation. This will help readers to understand how the arginine-rich peptides intercept via sequestering second messenger c-di-GMP signalling in P. aeruginosa and K. pneumoniae biofilms.
AR2: We have added a new paragraph in the introduction section explaining the role of c-di-GMP in forming a biofilm so that the reader could connect it better with the arginine-rich patterns/motifs encountered by our complex networks approach among the antibiofilm peptides. Please see changes in red font in the manuscript (lines 89-95). The paragraph is the following: “Other promising second-messenger-like target in bacteria is the cyclic diguanylate (c-di-GMP) which regulates a wide range of cellular functions from biofilm formation to growth and survival. In this sense, a c-di-GMP–sequestering peptide (CSP) was designed considering the high binding ability of the c-di-GMP with one of its protein effectors, a CheY-like (Cle) protein found in Caulobacter crescentus. The CSP was developed from a short arginine-rich region located at the C termini of Cle [5,20].”
RC3: L. 26. Change " threaten " to "threat. "
AR3: Corrected in the revised version
RC4: For clarity, I would suggest increasing the font size in Figure 4.
AR4: The figure 4 was improved. Font size was increased according to the importance of the nodes in the network
Reviewer 3 Report
Overall, this is a clear, concise, and well-written manuscript. The introduction is relevant and theory-based. Sufficient information about the previous study findings is presented for readers to follow the present study rationale and procedures. The methods are generally appropriate, and results and discussion are clear, and the conclusion is well-written. However, there is a list of questions that the authors should answer and include in the manuscript:
Abstract:
Overall, the abstract provides an interesting overview of the potential of antibiofilm peptides (ABFPs) as a basis for developing new antimicrobial agents. This is well-structured and uses technical language appropriately. However, there are some issues with clarity and coherence.
1. Benefit from more concise and precise language is missing.
2. Additionally, the paragraph would be easier to follow if the authors provided more background information about ABFPs and biofilms before delving into their research.
3. Finally, while the authors do an excellent job explaining the methodology they used, they could provide more detail on the results they obtained and what implications these results may have.
Introduction:
The introduction provides a good overview of the importance of biofilms and the challenges they pose to human health, while there have some lacunas
4. While the introduction mentions several ongoing efforts to develop antibiofilm agents, it does not provide a clear research gap or question the study aims to address.
5. Additionally, while machine learning and graph/network science tools are mentioned in the last paragraph, it is unclear how these will be used to address the research gap or question.
Materials and Methods:
1. The version and references in all the tools and soft wares are missing in the methodology section.
2. Authors can give 1 line of detail on the potential benefits and drawbacks of using METNs over other types of networks.
3. Authors don’t provide any information about how the optimal similarity threshold was determined beyond stating that network parameters were analyzed. This could make it difficult for others to replicate the study or understand the decision-making process.
4. It is unclear why the range of similarity thresholds tested was chosen (0 to 0.95). A justification for this choice would help to provide context for the analysis.
5. Authors mention that Gephi allows for calculating a "more comprehensive set of network parameters," but it does not explain why these particular parameters were chosen or how they are relevant to the research question.
6. Authors imply that the optimal similarity threshold will be selected based solely on network parameters without explaining how these parameters relate to the chemical space of the ABFPs or the study's overall goals.
7. How authors choose a 10% lower centrality cutoff for deleting the most central node in each metric.
8. In the methodology section, authors add the cutoff thresholds to reduce the ABFPs but on what basis they selected all these cutoffs is missing.
Results and Discussion:
The results are well-written and clearly explain the concepts and methods used. However, there have some questions and suggestions the authors should address:
1. The results lack clarity in explaining why the network density of HSPNs is much lower than the usually acceptable value of 0.1 in line 268.
2. The results do not discuss the potential impact of the similarity cutoff value on the biological interpretation of the network I line 277.
3. How does the degree distribution of HSPN change when the optimal similarity cutoff is applied?
4. The author could have provided more information about the practical implications of selecting an optimal similarity cutoff for constructing HSPNs of ABFPs in lines 283-304.
5. There is some direct sentence in the result and the discussion part needs to add reference. For example “The 384 starPep_00000, a 26 aa length ABFP that was derived from Melittin (bee venom) show up as a promising candidate since several pharmacological activities have been assigned to it besides the antibiofilm one, and has also been extensively evaluated against many targets (Ref).”
Conclusion:
The conclusion provides a comprehensive summary of the study, including the successful application of the Half-Space Proximal Networks to identify a subset of 66 ABFPs that represent the chemical space of the ABFPs. However further findings for future research and the potential limitations of the network-based approach in identifying novel antimicrobial agents are missing in the conclusion section. The conclusion should be more crispy. It would be helpful if the authors provided more specific details on the potential applications of the identified ABFPs and motifs in developing next-generation antimicrobials.
General comments:
1. Some sentences are quite long and could be broken down into smaller, more concise statements for better readability.
2. Add more discussion about how peptides help to stop the developing of the biofilm?
3. Provide a workflow in the methodology section that the authors used to enhance readers' comprehension.
4. There are grammatical mistakes throughout the manuscript. The authors should take care to correct them.
Author Response
Reviewer’s comments (RC1) Overall, this is a clear, concise, and well-written manuscript. The introduction is relevant and theory-based. Sufficient information about the previous study findings is presented for readers to follow the present study rationale and procedures. The methods are generally appropriate, and results and discussion are clear, and the conclusion is well-written. However, there is a list of questions that the authors should answer and include in the manuscript:
Abstract:
Overall, the abstract provides an interesting overview of the potential of antibiofilm peptides (ABFPs) as a basis for developing new antimicrobial agents. This is well-structured and uses technical language appropriately. However, there are some issues with clarity and coherence.
- Benefit from more concise and precise language is missing.
- Additionally, the paragraph would be easier to follow if the authors provided more background information about ABFPs and biofilms before delving into their research.
- Finally, while the authors do an excellent job explaining the methodology they used, they could provide more detail on the results they obtained and what implications these results may have.
Author’s Response (AR1): We acknowledge the reviewer’s comments that actually have improved the quality of the manuscript. We actually have fulfilled most of suggestions proposed by the reviewer in all sections of the paper. However, it is difficult to dedicate more space to provide further information on the antibiofilm peptides and biofilms in the Abstract, especially when the core of the article is the methodology (in what consist in? and why is developed for?). The results/conclusions presented in the abstract reflect exactly the described in the main text. On the other hand, the journal demands only 200 words maximum and other considerations states the following: “The abstract should be an objective representation of the article: it must not contain results which are not presented and substantiated in the main text and should not exaggerate the main conclusions”.
Introduction:
The introduction provides a good overview of the importance of biofilms and the challenges they pose to human health, while there have some lacunas
RC2: While the introduction mentions several ongoing efforts to develop antibiofilm agents, it does not provide a clear research gap or question the study aims to address.
AR2: The introduction section was overall improved by following your suggestions and by the other reviewers. Please see changes in red font at the manuscript (lines 80-105)
RC3: Additionally, while machine learning and graph/network science tools are mentioned in the last paragraph, it is unclear how these will be used to address the research gap or question.
AR3: This part was improved by rewriting the section corresponding to the lines 112-128.
Materials and Methods:
RC4: The version and references in all the tools and software are missing in the methodology section.
AR4: It was corrected in the revised version. Please see changes in red font at the corresponding section.
RC5: Authors can give 1 line of detail on the potential benefits and drawbacks of using METNs over other types of networks.
AR5: It was corrected in the revised version. Please see changes in red font at the corresponding section (lines 174-177)
RC6: Authors don’t provide any information about how the optimal similarity threshold was determined beyond stating that network parameters were analysed. This could make it difficult for others to replicate the study or understand the decision-making process.
AR6: A detailed explanation of how the optimal similarity cutoff was determined is explained in the first section of Results and Discussion (particular attention to lines 348-360).
RC7: It is unclear why the range of similarity thresholds tested was chosen (0 to 0.95). A justification for this choice would help to provide context for the analysis.
AR7: A justification for the selection of the similarity cutoff range is given at the corresponding section of Mat and Met. Please see changes in red font at the manuscript.
RC8: Authors mention that Gephi allows for calculating a "more comprehensive set of network parameters," but it does not explain why these particular parameters were chosen or how they are relevant to the research question.
AR8: The idea was clarified in the corresponding section. Please see changes in red font at the corresponding section of the MS
RC9: Authors imply that the optimal similarity threshold will be selected based solely on network parameters without explaining how these parameters relate to the chemical space of the ABFPs or the study's overall goals.
AR9: This part was improved by stating that the selection of the optimal cutoff was also accompanied by visual mining of the resulting networks topology. Please see changes in red font at the corresponding section of the MS. (Please also consult File 1SM that also includes the GraphML files).
RC10: How authors choose a 10% lower centrality cutoff for deleting the most central node in each metric.
AR10: this approach is unnecessary and was removed in the current version of the paper.
RC11: In the methodology section, authors add the cutoff thresholds to reduce the ABFPs but on what basis they selected all these cutoffs is missing.
AR11: The explanation on why different cutoffs were selected to reduce the ABFPs was improved in the subsection 2.5. Please the changes in red font at the corresponding section of the MS (lines 218-227)
Results and Discussion:
The results are well-written and clearly explain the concepts and methods used. However, there have some questions and suggestions the authors should address:
RC12: The results lack clarity in explaining why the network density of HSPNs is much lower than the usually acceptable value of 0.1 in line 268.
AR12: We have improved the explanation on why the network density of HSPNs is much lower than classical similarity networks. Please the changes in red font at the corresponding section (lines 324-331)
RC13: The results do not discuss the potential impact of the similarity cutoff value on the biological interpretation of the network I line 277.
AR13: We added a new paragraph introducing the potentialities of the HSPN with optimal cutoff, that will be addressed from this point on. Please the changes in red font at the corresponding section (lines 355-359)
RC14: How does the degree distribution of HSPN change when the optimal similarity cutoff is applied?
AR14: The redaction of this part was improved in the manuscript. Please the changes in red font at the corresponding section (lines 360-369)
RC15: The author could have provided more information about the practical implications of selecting an optimal similarity cutoff for constructing HSPNs of ABFPs in lines 283-304.
AR15: We added a new paragraph introducing the potentialities of the HSPN with optimal cutoff, that will be addressed from this point on. Please the changes in red font at the corresponding section (lines 355-359)
RC16: There is some direct sentence in the result and the discussion part needs to add reference. For example, “The 384 starPep_00000, a 26 aa length ABFP that was derived from Melittin (bee venom) show up as a promising candidate since several pharmacological activities have been assigned to it besides the antibiofilm one, and has also been extensively evaluated against many targets (Ref).”
AR16: Two references [54, 55] were inserted at end of this sentence to support it
Conclusion:
RC17: The conclusion provides a comprehensive summary of the study, including the successful application of the Half-Space Proximal Networks to identify a subset of 66 ABFPs that represent the chemical space of the ABFPs. However further findings for future research and the potential limitations of the network-based approach in identifying novel antimicrobial agents are missing in the conclusion section. The conclusion should be crispier. It would be helpful if the authors provided more specific details on the potential applications of the identified ABFPs and motifs in developing next-generation antimicrobials.
AR17: Many thanks for your recommendation. We believe the presented methodology is emerging and many of its further potentialities will be discovered with the time being. We have introduced two paragraphs in the “Conclusion” section that could address your suggestion. Please see changes in red font. Furthermore, along the “Results and Discussion” section we have highlighted the relevance of the 66 ABFPs and motifs for the discovery/design of next-generation antimicrobials
General comments:
RC18: Some sentences are quite long and could be broken down into smaller, more concise statements for better readability.
AR18: We have improved the writing along the manuscript by considering your suggestions and the others made by the other reviewers. Please see changes in red font along the MS.
RC19: Add more discussion about how peptides help to stop the developing of the biofilm?
AR19: This information on how peptides help stopping/eradicating biofilms is summarized in Table 4. The 14 types of action mechanisms for stopping/eradicating biofilms represented by each one of the peptides listed in Table 4 were mapped in the proposed representative antibiofilm chemical space. From this mapping, we realized the ABFP chemical space represented by the HSPN encompassed all the 14 classes of antibiofilm action modes. Considering the reduced size of the HSPN (66 ABFPs), 9 out of the 14 antibiofilm action modes were seamless connected (mapped) with HSPN nodes while the remaining 5 were connected with HSPN nodes but at weaker similarities (<0.60). Analyses on the distribution of these 14 representative peptides among the HSPN communities is also presented (section 3.4). On the other hand, additionally information on the role of c-di-GMP as putative target to stop biofilm formation was included in the Introduction.
RC20: Provide a workflow in the methodology section that the authors used to enhance readers' comprehension.
AR20: The subsection 2.8. “Overall Workflow Integrating Complex Networks to Next-Generation Antimicrobials Development” was added to the Materials and Methods section along with the corresponding Figure 1 representing the methodological workflow.
RC21: There are grammatical mistakes throughout the manuscript. The authors should take care to correct them.
AR21: They were corrected along the revision process, thanks